# Exploring the Role of Oleic Acid in Muscle Cell Differentiation: Mechanisms and Implications for Myogenesis and Metabolic Regulation in C2C12 Myoblasts

**DOI:** 10.3390/biomedicines13071568

**Published:** 2025-06-26

**Authors:** Francesco Vari, Elisa Bisconti, Ilaria Serra, Eleonora Stanca, Marzia Friuli, Daniele Vergara, Anna Maria Giudetti

**Affiliations:** 1Department of Physiology and Pharmacology, Sapienza University of Rome, 00185 Rome, Italy; francesco.vari@uniroma1.it (F.V.); ilaria.serra@uniroma1.it (I.S.); marzia.friuli@uniroma1.it (M.F.); 2Department of Biological and Environmental Sciences and Technologies, University of Salento, 73100 Lecce, Italy; elisa.bisconti@studenti.unisalento.it (E.B.); anna.giudetti@unisalento.it (A.M.G.); 3Consorzio Interuniversitario Nazionale per la Scienza e Tecnologia dei Materiali, 50121 Firenze, Italy; 4Department of Experimental Medicine, University of Salento, 73100 Lecce, Italy; eleonora.stanca@unisalento.it

**Keywords:** C2C12 myoblast differentiation, glucose uptake, lipid droplets, oleic acid, skeletal muscle

## Abstract

**Background/Objectives**: Myogenesis, the process by which myoblasts differentiate into multinucleated muscle fibers, is tightly regulated by transcription factors, signaling pathways, and metabolic cues. Among these, fatty acids have emerged as key regulators beyond their traditional role as energy substrates. Oleic acid, a monounsaturated fatty acid, has been shown to modulate muscle differentiation, potentially influencing myogenic pathways. This study examines the role of oleic acid in promoting C2C12 myoblast differentiation and its associated molecular mechanisms, comparing it to standard horse serum (HS)-based differentiation protocols. **Methods**: C2C12 murine myoblasts were cultured under proliferative conditions and differentiated using DMEM supplemented with either 2% HS or oleic acid (C18:1, n-9). The molecular signaling pathway was evaluated by measuring the expression of p38 MAPK, β-catenin, GLUT4, and NDRG1. **Results**: Oleic acid promoted the differentiation of C2C12 cells, as evidenced by a progressively elongated morphology, as well as the induction of muscle-specific myogenin, myosin heavy chain (MHC), and *MyoD*. Moreover, oleic acid reduced the expression of Atrogin-1 and *MuRF1* ubiquitin E3 ligase. BODIPY staining revealed the enhanced accumulation of lipid droplets in oleic acid-treated cells. The Western blot analysis demonstrated robust activation of p38 MAPK and β-catenin pathways in response to oleic acid, compared with HS. Additionally, oleic acid upregulated GLUT4 expression and increased the phosphorylation of insulin receptor and NDRG1, indicating an enhanced glucose uptake capacity. **Conclusions**: These findings demonstrate that oleic acid promotes C2C12 myoblast differentiation and improves glucose uptake via GLUT4. Oleic acid emerges as a promising metabolic regulator of myogenesis, offering potential therapeutic applications for muscle regeneration in muscle-related pathologies.

## 1. Introduction

Myogenesis, the process by which progenitor cells differentiate into mature, multinucleated muscle fibers, is a highly regulated biological program comprising sequential phases of proliferation, lineage commitment, fusion, and maturation. This complex differentiation cascade is orchestrated by a tightly controlled network of transcription factors, signaling pathways, and extracellular cues that together ensure the proper development and functionality of skeletal muscle [1,2].

In recent years, alongside canonical regulators of myogenesis, such as the MyoD family of transcription factors, increasing attention has been paid to the role of metabolic and endocrine factors in modulating muscle development. Fatty acids have emerged as influential players in muscle biology, not only as metabolic substrates but also as bioactive signaling molecules capable of regulating satellite cell activation, migration, and differentiation [3,4]. These effects are particularly evident in the context of skeletal muscle regeneration, where fatty acid composition and availability can modulate repair efficiency under both physiological and pathological conditions.

Among various lipid species, omega-3 polyunsaturated fatty acids have been widely studied for their ability to reduce inflammation and promote muscle regeneration, presenting promising therapeutic potential for sarcopenia, cachexia, and muscular dystrophies [5,6]. Moreover, the local inflammatory milieu, shaped in part by lipid-derived mediators such as eicosanoids, can significantly influence myogenesis by modulating the balance between pro- and anti-inflammatory signaling pathways [7]. Importantly, the dietary ratio of omega-6 to omega-3 fatty acids affects not only systemic inflammation but also muscle health, strength, and longevity [8].

Within this broader lipid landscape, oleic acid, a monounsaturated omega-9 fatty acid and a major constituent of olive oil, has garnered increasing interest due to its anti-inflammatory, insulin-sensitizing, and mitochondria-boosting properties [9,10,11]. Beyond its energetic role, oleic acid acts as a potent regulator of intracellular signaling pathways. It has been shown to modulate gene expression, improve mitochondrial biogenesis, and influence key regulators of muscle metabolism and differentiation, including peroxisome proliferator-activated receptors and AMPK [12,13].

Studies have also highlighted oleic acid’s capacity to alleviate endoplasmic reticulum stress and enhance insulin sensitivity in skeletal muscle cells, effects mediated in part through the activation of the Akt/protein kinase B pathway, which promotes glucose uptake via GLUT4 translocation and supports myoblast differentiation through mTOR signaling and the inhibition of atrophy-related genes such as *MuRF1* and *MAFbx* [13,14,15]. Furthermore, the activation of p38 mitogen-activated protein kinase (p38MAPK) has been identified as a key driver of myogenic commitment and fusion, working upstream of *MyoD* and myogenin during muscle cell differentiation [16]. Within this pathway, N-myc downstream-regulated gene 1 (NDRG1) has emerged as a downstream effector of p38MAPK, playing critical roles in cell cycle arrest, cytoskeletal organization, and lineage specification [17]. The phosphorylation of NDRG1 has been linked to an enhanced differentiation capacity and may serve as a convergence point between metabolic and stress-responsive signaling in muscle cells.

In addition to these signaling pathways, lipid droplets (LDs), traditionally viewed as passive lipid storage sites, have recently been recognized as active participants in myogenesis. LDs are composed of a core of neutral lipids, primarily triacylglycerols (TAGs) and cholesteryl esters (CEs), surrounded by a phospholipid monolayer and associated proteins. Beyond their role in energy storage, LDs interact with various cellular organelles and are involved in lipid metabolism, signaling, and membrane trafficking.

Emerging evidence suggests that LDs contribute to myoblast differentiation by facilitating cytoskeletal remodeling necessary for cell migration and fusion. In C2C12 myoblasts, an increased LD content has been associated with the enhanced formation of multinucleated myotubes. Mechanistically, LD-associated proteins such as acyl-CoA synthetase long-chain family member 3 and lysophosphatidylcholine acyltransferase 1 have been implicated in recruiting actinin proteins to LD surfaces, promoting actin filament formation and remodeling essential for myogenic progression [18]. Additionally, LD dynamics have been linked to satellite cell fate decisions, where differential LD accumulation influences the balance between self-renewal and differentiation, highlighting the multifaceted roles of LDs in muscle development and regeneration [19].

Despite growing evidence of the beneficial effects of oleic acid on metabolic health and mitochondrial efficiency, especially when compared to saturated fatty acids, its precise role in orchestrating the molecular events underlying myoblast differentiation remains incompletely defined.

In this study, we investigated the effects of oleic acid on myogenic differentiation using the murine C2C12 myoblast cell line, a well-established in vitro model of skeletal muscle development and regeneration, exhibiting a myogenic program highly comparable to human satellite cells [20,21]. We compared the effects of oleic acid to those induced by horse serum (HS), a commonly used culture supplement that promotes myogenesis through a complex mixture of growth factors and hormones [20,22]. Notably, oleic acid enhanced myogenic differentiation more effectively than other unsaturated fatty acids tested. Mechanistically, oleic acid promoted differentiation through the activation of the p38MAPK/NDRG1 axis, independently of the broader trophic effects of HS, and was associated with a reduction in catabolic gene expression. These findings underscore the unique signaling properties of oleic acid in muscle cells and shed light on its potential application in strategies aimed at promoting muscle regeneration and treating muscle-wasting disorders.

## 2. Materials and Methods

### 2.1. Cell Cultures

The murine myoblast cell line C2C12 (from ATCC^®^ CRL-1772™) was used for all experiments. Cells were cultured in high-glucose Dulbecco’s Modified Eagle’s Medium (DMEM; D5796, Sigma-Aldrich, St. Louis, MO, USA), supplemented with 10% fetal bovine serum (FBS; HI-12A, Capricorn, Ebsdorfergrund, Germany), 100 μg/mL streptomycin, and 100 U/mL penicillin (#P4333, Sigma-Aldrich, St. Louis, MO, USA). Fatty acids were conjugated to fatty-acid-free bovine serum albumin (BSA; 03117057001, Sigma-Aldrich, St. Louis, MO, USA) at a final molar ratio of 2:1 (fatty acid to BSA), as described in [23], which approximates the ratio found in human serum, mimicking their physiological transport [24]. The mixture was then diluted in DMEM to achieve a final fatty acid concentration of 200 µM.

### 2.2. Differentiation Protocol and Myotubes Quantification

C2C12 cells were seeded in 6-well plates at a density of 5 × 10^5^ cells per well with high-glucose DMEM supplemented with 10% FBS and antibiotic solution. Once cell confluency reached 90–100%, the cultures were gently washed twice with phosphate-buffered saline (PBS; D8537, Sigma-Aldrich, St. Louis, MO, USA) and subsequently switched to differentiation medium. The differentiation medium consisted of DMEM supplemented with either 100 µM BSA (control), 2% HS (23G306, Sigma-Aldrich, St. Louis, MO, USA), or 200 µM of oleic acid (C18:1, n-9 COD. 75090), linoleic acid (C18:2, n-6 COD. 62230), α-linolenic acid (C18:3, n-3 COD. 62160), and arachidonic acid (C20:4, n-6 COD. 6610-25-9), in the absence of 10% FBS. Cells were then maintained under these conditions for varying durations to allow for myogenic differentiation. On the third day of differentiation, the number of multinucleated myotubes was quantified by counting the myotubes present within a standardized field of view (1 mm^2^ × 100), using a phase-contrast microscope (AMEX100, EVOS™ XL Core Imaging System, Thermo Fisher Scientific, Waltham, MA, USA), as described [25].

### 2.3. Viability Test

Cell viability on differentiating C2C12 cells was assessed using the non-toxic Alamar Blue reagent (DAL1025, Thermo Fisher Scientific, Waltham, MA, USA). On the day of the assay, the reagent was added directly to the culture medium, and cells were incubated for 4 h at 37 °C. After incubation, the reagent was removed and replaced with fresh medium, allowing cells to resume normal proliferation. Absorbance was measured at 570 nm at each time point using a Biotek Cytation 5 Cell Imaging Multimode Reader (Agilent Technologies, Santa Clara, CA, USA).

### 2.4. Cell Lysis and Western Blotting Analysis

Proteins were extracted from cells using RIPA lysis buffer (9806, Cell Signaling, Danvers, MA, USA). The Bradford method measured total protein levels (Bio-Rad Laboratories, Hercules, CA, USA). Proteins were separated by SDS-polyacrylamide gel electrophoresis and transferred to a nitrocellulose membrane (Bio-Rad Laboratories, Hercules, CA, USA). After blocking, at room temperature for 1 h with 5% (*w*/*v*) skim milk in TBS-Tris buffer (Tris-buffered saline (TBS) plus 0.5% (*v*/*v*) Tween-20, TTBS), membranes were incubated with primary antibodies against phosphoinositide-dependent kinase-1 (PDK1, #3062, Cell Signaling Technology, Danvers, MA, USA), phospho-PDK1 (pPDK, #3061, Cell Signaling Technology, Danvers, MA, USA), NDRG1 (Ab124713, Abcam, Cambridge, UK), phospho-NDRG1 (pNDRG1, ab124713, Abcam, Cambridge, UK), myosin heavy chain 4 (MyHC4, 14-6503-82 eBioscience, Thermo Fisher Scientific, Waltham, MA, USA), p38MAPK (#9212, Cell Signaling Technology, Danvers, MA, USA), insulin receptor (IR, GTX101136 GeneTex, Irvine, CA, USA), phospho-IR (pIR, SC81500, Santa Cruz Biotechnology, Dallas, TX, USA), GLUT4 (#2213, Cell Signaling Technology, Danvers, MA, USA), and β-catenin (610154, eBioscience, Thermo Fisher Scientific, Waltham, MA, USA). After washing with TTBS, blots were incubated with peroxidase-conjugated secondary antibodies (cat no A3687 and cat no A3652, Sigma-Aldrich, St. Louis, MO, USA) at 1:10,000 dilutions at room temperature for 1–2 h. The filters were then washed twice in TTBS. Western blotting analyses were performed using the Amersham ECL Advance Western Blotting Detection Kit (RPN2106, GE Healthcare, Little Chalfont, UK). A densitometric analysis of immunoblots was performed using Image LabTM software version 6.0.1 2017 (Bio-Rad Laboratories, Hercules, CA, USA). Red Ponceau staining was used as the protein loading control.

### 2.5. BODIPY (493/503) Staining of LD

For BODIPY staining, after treatments, C2C12 cells were incubated for 15 min at 37 °C with BODIPY 493/503 at a dilution of 1:1000 in PBS from a stock solution of 1 mg/mL. Finally, cells were stained with DAPI for 10 min, washed three times, images were captured using a Cell Imaging Station microscope (Invitrogen™, FLoid™ Cell Imaging Station, 10-213-412, Thermo Fisher Scientific, Waltham, MA, USA) at 20× magnification with a scale bar of 100 μm. Fluorescence intensity was measured by a Biotek Cytation 5 Cell Imaging Multimode Reader (Agilent Technologies, Santa Clara, CA, USA).

### 2.6. Thin-Layer Chromatography (TLC) Analysis of Lipids

Total lipids were extracted using methyl-tert-butyl ether as an extraction solvent, as reported in [25]. Extracted lipids were loaded on silica gel plates for TLC analysis. Plates were developed with hexane/ethyl ether/acetic acid (70/30/1; *v*/*v*/*v*) for neutral lipid separation. After development, plates were uniformly sprayed with 10% cupric sulfate in 8% aqueous phosphoric acid, allowed to dry for 10 min at room temperature, and then placed into a 145 °C oven for 10 min, as reported in [26]. Different lipid species were identified by developing specific standards under the same experimental conditions. The densitometric analysis of spot intensity was performed by the ChemiDoc system (Bio-Rad).

### 2.7. RNA Extraction and RT-qPCR Analyses

Total RNA was extracted with TRI reagent^®^ (R2050-1, Zymo Research, R2050-1, Irvine, CA, USA). RNA quantification was performed using a NanoDrop One Spectrophotometer (Thermo Fisher Scientific, Waltham, MA, USA). Gene expression analyses were performed through real-time PCR using SYBR Green technology (FluoCycle, Euroclone, Pero, Italy), run on a CFX Connect Real-time System (Bio-Rad Laboratories, Hercules, CA, USA). Glyceraldehyde-3-phosphate dehydrogenase (GAPDH) was used as a housekeeping gene, and the relative expression levels were expressed as fold change. The list of primers is reported on Appendix A.

### 2.8. Statistical Analysis

Results are expressed as mean ± standard error (SE). Statistical differences were assessed using GraphPad Prism version 8.3.0 for Windows. Comparisons were performed using a one-way analysis of variance (ANOVA) or Student’s *t*-test, where applicable. After Tukey’s post hoc analysis, differences between groups were considered statistically significant when *p* < 0.05.

## 3. Results

### 3.1. Effect of Fatty Acids on C2C12 Myoblast Differentiation

Fatty acids can influence myoblast fusion efficiency and muscle fibers’ maturation, impacting myotube morphology and functionality [3,4]. We monitored the proliferation and differentiation process of C2C12 cells over three consecutive days while treating the cells with oleic, linoleic, linolenic, and arachidonic acid, compared with the standard differentiation protocol, which included HS.

Differentiation was initiated once the myoblasts reached full confluence. As differentiation progressed, the cells underwent a morphological transformation, from a rounded form to elongated, tubular structures due to cell fusion. After serum starvation, the myogenic program was induced by HS or fatty acids and monitored over days 1, 2, and 3 via phase-contrast microscopy (Appendix A). As differentiation progressed, cells gradually adopted a tubular morphology and began aligning with neighboring cells. By day 3, phase-contrast images revealed distinct myotube morphologies depending on the treatment, with oleic acid showing particularly robust effects (Figure 1A). Higher-resolution microscopic scans are provided for the treatments with HS, BSA, and oleic acid, allowing for a clearer visualization of myotube morphology (Appendix A). The quantification of myotube formation at day 3 is shown in the bar graph (Figure 1B), illustrating the number of myotubes per microscopic field (expressed as fold change) across the various treatments. Both HS and oleic acid significantly increased the number of myotubes per field compared to the BSA, indicating enhanced myogenic differentiation. The levels induced by oleic acid were comparable to those observed with HS, suggesting that oleic acid can effectively mimic the differentiation-promoting effects of serum. In contrast, other fatty acids such as linoleic acid and α-linolenic acid induced significantly fewer myotubes, highlighting a specific role for oleic acid in promoting myotube formation. The conditions did not induce cell death over the three days, except for arachidonic acid, which caused a significant increase in cell death starting from the second day of incubation (Figure 1C).

### 3.2. Oleic Acid Induces Myoblast Differentiation

MyHC4 is a well-established marker of C2C12 myoblast differentiation, reflecting the transition from proliferative myoblasts to mature, myotubes [27]. To evaluate the differentiation status under various treatments, MyHC4 protein expression was assessed by Western blotting. As shown in Figure 2A, cells cultured with BSA for three days did not display MyHC4 expression, consistent with an undifferentiated state. In contrast, HS-treated cells exhibited clear myotube formation by day 3 (Figure 1A,B), with the corresponding upregulation of MyHC4 (Figure 2A), indicating that HS effectively promotes myogenic differentiation, likely due to its rich content of growth factors and hormones [28].

Among the fatty acids tested, oleic acid induced a pronounced increase in MyHC4 expression, higher than that observed in HS-treated cells, as well as in those treated with other fatty acids. Arachidonic acid, on the other hand, failed to elicit detectable MyHC4 expression (Figure 2A), suggesting a potentially inhibitory or neutral role in differentiation, consistent with previous findings linking high levels of arachidonic acid to inflammatory signaling and catabolic pathways [29].

Compared to BSA-treated cells, oleic acid treatment also enhanced the mRNA level of myogenic regulatory factors *MyoD* and myogenin (*MyoG*) (Figure 2B), which are critical markers of early and late stages of differentiation, respectively. *MyoD* initiates the myogenic program by activating muscle-specific gene expression, while *MyoG* is essential for terminal differentiation and myotube formation [30]. However, the effect was less pronounced than that induced by HS on the same genes.

In addition, we investigated the expression of two muscle-specific E3 ubiquitin ligases, *MuRF1* and *MAFbx* (also known as Atrogin-1), both of which are upregulated during muscle atrophy and contribute to the proteasomal degradation of muscle proteins [17]. Treatment with oleic acid resulted in a moderate reduction in *MuRF1* expression compared to the BSA, while HS significantly increased *MuRF1* levels, indicating a potential pro-catabolic response (Figure 2C). A similar trend was observed for *MAFbx*, with oleic acid reducing its expression (Figure 2B). In this case, HS did not significantly affect *MAFbx* mRNA expression, although there was a trend toward a decrease. These findings suggest that oleic acid not only supports differentiation but may also confer anti-catabolic effects [31,32].

Overall, these data support the hypothesis that oleic acid promotes myogenic differentiation and protects against muscle atrophy, possibly through a dual mechanism of enhancing myogenic gene expression while suppressing key atrogenes. In contrast, HS appears to induce a broader activation of both anabolic and catabolic processes, reflecting the complex milieu of signaling molecules present in serum that may simultaneously promote growth and remodeling under in vitro conditions.

### 3.3. Fatty Acids Differently Induce LD Accumulation During C2C12 Differentiation

Fatty acids entering skeletal muscle cells can be incorporated into LDs. These structures not only act as critical energy reservoirs but also contribute to cellular lipid homeostasis and signaling during myoblast differentiation [33]. Emerging evidence suggests that LDs are not merely passive storage sites but may actively participate in the regulation of myogenesis, possibly by modulating lipid signaling, energy availability, and membrane remodeling required for myotube formation [18].

To determine whether the observed differences in fatty acid-induced differentiation correlated with alterations in LD dynamics, we performed BODIPY staining on C2C12 cells after three days of treatment with various fatty acids. Fluorescence microscopy revealed distinct patterns of LD accumulation depending on the fatty acid provided (Figure 3A,B). Notably, oleic acid induced the most robust accumulation of LDs, consistent with its known ability to promote TAG synthesis and LD formation via the activation of enzymes such as DGAT1 and DGAT2 [34]. This extensive LD accumulation may provide a readily accessible energy reservoir and buffering system for membrane lipids and signaling intermediates, supporting the energy-intensive process of muscle differentiation.

In contrast, cells treated with linoleic (18:2) and α-linolenic acid (18:3) exhibited moderate LD accumulation, while arachidonic acid (20:4) induced minimal LD accumulation, suggesting that their incorporation into LDs is less efficient than oleic acid (Figure 3A,B).

To validate these microscopy-based observations, we performed TLC to quantify the intracellular lipid classes after three days of fatty acid treatment (Figure 3C). Consistent with the fluorescence data, all fatty acids increased TAG accumulation, though to varying degrees. Oleic acid again led to the most substantial TAG enrichment, while arachidonic acid resulted in the lowest TAG levels among the tested fatty acids (Figure 3C,D). These results reinforce the idea that oleic acid preferentially promotes neutral lipid storage, possibly creating a more favorable metabolic environment for myogenic progression.

Interestingly, cells cultured in the presence of HS showed no significant increase in TAGs relative to control cells but instead demonstrated a notable enrichment in CE. This likely reflects the complex lipid profile of serum, which includes high levels of lipoprotein-bound cholesterol and esterified lipids.

Across all fatty acid treatments, a reduction in cholesterol content was observed, with oleic and linoleic acid treatments producing the most marked decreases. This could suggest a shift in lipid metabolism favoring TAG over cholesterol synthesis, potentially due to substrate competition or selective enzyme regulation. No significant changes were detected in free fatty acid (FFA) or diacylglycerol (DAG) levels among the treatment groups, indicating that the major impact of fatty acid supplementation was specifically directed toward TAG and cholesterol pools (Figure 3C,D).

Together, these results suggest that oleic acid facilitates myoblast differentiation not only through its direct signaling properties but also by remodeling intracellular lipid stores, promoting TAG-rich LD formation and reducing cholesterol accumulation. Such lipid remodeling may optimize the cellular lipid environment to support membrane biogenesis, energy production, and the activation of myogenic signaling pathways.

### 3.4. Myoblast Differentiation with Fatty Acids Activates the p38 MAPK Pathway

To elucidate the molecular mechanisms by which oleic acid promotes myogenic differentiation in C2C12 cells, we focused on identifying the intracellular signaling pathways activated in response to oleic acid treatment and compared them with those induced by HS. Given the well-established role of the p38 MAPK pathway in regulating muscle cell differentiation, particularly through its involvement in myogenic transcription factor activation and myoblast fusion [35], we hypothesized that oleic acid may differentially engage this pathway compared to HS.

The Western blot analysis of C2C12 myoblasts treated for three days with oleic acid or HS revealed a significant increase in p38 MAPK protein expression in oleic acid-treated cells (Figure 4A,B). This suggests that oleic acid may exert its pro-differentiation effects at least in part through the potentiation of p38 MAPK signaling. During skeletal muscle differentiation, both the p38 MAPK and Wnt/β-catenin signaling pathways play essential roles in coordinating the transition from proliferating myoblasts to differentiated myotubes. Recent studies have highlighted a functional interaction between p38 MAPK and β-catenin [36], a dual-function protein that regulates cell–cell adhesion and acts as a central effector in the Wnt signaling pathway, both of which are vital for the progression of myoblast differentiation [37,38]. To explore this aspect, we assessed the expression of β-catenin. The Western blot analysis demonstrated that β-catenin levels were substantially elevated in oleic acid-treated cells compared to those treated with HS (Figure 4A,C), reinforcing the idea that oleic acid activates a distinct pro-differentiation axis involving the p38 MAPK/β-catenin cascade.

Taken together, these findings suggest that oleic acid not only promotes myogenic differentiation through classical energy-providing and metabolic mechanisms but also activates a specialized signaling network. This includes the enhanced activation of p38 MAPK and upregulation of β-catenin, collectively supporting a more robust and possibly qualitatively distinct differentiation response compared to traditional serum-based methods.

### 3.5. Oleic Acid Influences Myoblast Differentiation by Targeting Insulin Signaling

One of the key downstream targets of the p38 MAPK pathway during myoblast differentiation is GLUT4, an insulin-responsive transporter critical for glucose uptake in skeletal muscle cells [39]. The regulation of GLUT4 is essential not only for efficient glucose handling but also for maintaining overall muscle energy homeostasis and functionality. Given the importance of GLUT4 in muscle metabolism, we investigated the effects of oleic acid and HS treatments on insulin signaling and GLUT4 expression in C2C12 cells.

Our results showed a marked upregulation of GLUT4 expression in response to oleic acid treatment when compared to cells exposed to HS (Figure 5A,D), suggesting that oleic acid may enhance the insulin sensitivity of muscle cells. To gain deeper insight into the insulin signaling pathway, we examined the expression and activation status of IR, assessing both total IR and its phosphorylated form (pIR). The phosphorylation of IR represents a critical early event in insulin signaling, as it initiates a cascade of downstream molecular interactions that culminate in glucose uptake. Notably, oleic acid treatment significantly increased pIR levels, indicating a more robust activation of the insulin receptor compared to HS treatment (Figure 5C).

To further characterize the downstream signaling events, we analyzed the expression of PDK1, a key effector required for the translocation of GLUT4 to the plasma membrane. The enhanced phosphorylation of PDK1 was observed in oleic acid-treated cells, confirming that the insulin signaling cascade was more effectively engaged under these conditions (Figure 5A,B).

NDRG1, a stress-responsive protein, is a well-established downstream target of insulin. NDRG1 has been shown to influence key aspects of myogenic differentiation, including cell morphology and membrane remodeling, which are critical for myoblast fusion and the formation of multinucleated myotubes [40,41]. Moreover, the expression of NDRG1 is regulated by p38MAPK [17,42,43]. Notably, we observed a marked increase in the phosphorylated-to-total NDRG1 ratio in oleic acid-treated cells compared to those exposed to HS (Figure 5A,E), indicating that NDRG1 is a significant downstream effector of oleic acid-induced insulin signaling activation. Collectively, these findings demonstrate that oleic acid enhances insulin-mediated signaling and promotes GLUT4 expression, thereby facilitating glucose uptake.

## 4. Discussion

The differentiation of myoblasts into mature myotubes represents a critical process in muscle development and regeneration. Our study demonstrates that fatty acids, particularly oleic acid, significantly influence myoblast fusion efficiency and muscle fiber maturation. These findings are consistent with previous reports indicating that fatty acids are essential in skeletal muscle differentiation and energy homeostasis [3,4]. A hallmark of successful myoblast differentiation is the formation of multinucleated myotubes, characterized by the expression of MyHC4.

The murine C2C12 cell line is a well-established in vitro model for studying myogenesis and muscle regeneration due to its strong ability to differentiate into multinucleated myotubes and activate a gene expression program closely resembling those of human satellite cells [20,44], thereby ensuring strong functional homology despite species differences. Its ease of genetic manipulation, rapid fusion kinetics, and high reproducibility make C2C12 cells ideal for evaluating bioactive compounds, especially lipids and fatty acids, in muscle regeneration contexts [45]. Numerous studies support its predictive value for lipid-induced myogenic responses relevant to human muscle repair [46].

In our study, C2C12 cells treated with different fatty acids exhibited varying myotube morphologies, with oleic acid being the most effective in promoting myoblast fusion. MyHC4 expression was significantly higher in oleic acid-treated cells than in those subjected to the standard HS differentiation protocol. This result aligns with prior findings identifying MyHC expression as a key marker of myogenic maturation and fiber-type specification [47].

In our experiments, we observed that oleic acid, analogously to HS, promotes the expression of the myogenic regulatory factors *MyoD* and myogenin (*MyoG*), suggesting that both treatments favor the activation of the myogenic program.

However, the two treatments showed opposing effects on the expression of atrophy-related genes. Specifically, oleic acid decreased the expression of *MuRF1* (Trim63) and *MAFbx* (Fbxo32), two muscle-specific E3 ubiquitin ligases that are key markers and mediators of muscle atrophy. In contrast, the HS treatment led to an increase in the expression of both *MuRF1* and *MAFbx*.

These divergent effects may be explained by the distinct biochemical compositions and signaling properties of oleic acid versus HS. Oleic acid, a monounsaturated fatty acid, has been shown to exert anti-inflammatory and anabolic effects on skeletal muscle. Studies report that oleic acid can activate insulin signaling [48], which not only promotes muscle protein synthesis but also suppresses the transcription of *MuRF1* and *MAFbx* via the inhibition of FoxO transcription factors [49]. Thus, oleic acid may directly downregulate muscle atrophy genes through Akt-mediated FoxO inhibition, while simultaneously promoting differentiation. On the other hand, HS, although commonly used to induce differentiation in myoblasts (e.g., in C2C12 cells), contains a complex mixture of growth factors, cytokines, and serum-derived components, including glucocorticoids and pro-inflammatory mediators, that may unintentionally activate catabolic pathways. Indeed, it has been shown that serum from adult animals, such as HS, can contain factors that upregulate atrogenes under certain conditions, possibly through the activation of NF-κB or FoxO pathways in stress-prone or sub-optimally differentiated cells [31,50]. Thus, the concurrent upregulation of *MuRF1* and *MAFbx* by HS may reflect a stress response or an incomplete differentiation state where catabolic signals remain active. This dual effect might mirror a complex in vitro environment where differentiation and atrophy-related pathways are both transiently active.

LDs have been reported to play an active role in myoblast differentiation, functioning not only as energy reservoirs but also as structural scaffolds that facilitate myofiber anchoring during this process [18]. This indicates that lipid storage is not merely a passive event but contributes dynamically to cellular remodeling and differentiation. In our analysis, we found that fatty acids differentially modulate the accumulation of LD and TAG, with oleic acid being the most effective in promoting LD formation. This observation aligns with the increased expression of MyHC4 detected in oleic acid-treated myocytes, suggesting enhanced myogenic progression under this condition.

Arachidonic acid did not accumulate significantly in LD, suggesting that it may be preferentially directed toward eicosanoid synthesis or other metabolic fates rather than storage. This finding is consistent with the known selectivity of cPLA2α, an enzyme involved in eicosanoid synthesis, for hydrolyzing phospholipids containing arachidonic acid at the sn-2 position [51]. However, an alternative explanation for the minimal LD accumulation observed with arachidonic acid treatment could be its well-documented lipotoxicity. Arachidonic acid exhibits cytotoxicity in vitro at concentrations overlapping with physiological ranges [52] and can induce apoptosis in various cell lines, including colon cancer cells [53]. Consequently, the reduced lipid storage observed in our experiments may result from cellular cytotoxic responses rather than, or in combination with, preferential metabolic utilization for eicosanoid synthesis.

Interestingly, oleic acids led to a more pronounced reduction in total cholesterol content within cellular lipids. Given that membrane fluidity is a critical factor for membrane fusion events during myogenesis, and that unsaturated fatty acids are known to reduce membrane rigidity and promote myoblast differentiation [54], we propose that oleic acid incorporation into membrane phospholipids enhances membrane dynamics, thereby creating a more permissive environment for myogenic fusion and maturation. Supporting this hypothesis, it has been reported that the fatty acid composition of differentiated myotube membranes is enriched in n-9 monounsaturated fatty acids, particularly oleic acid (18:1 n-9), within the glycerophospholipid fraction. This enrichment not only reflects the preferential incorporation of oleic acid into membrane structures during differentiation but also reinforces the idea that oleic acid plays a functional and regulatory role in the myogenic process.

Our study revealed that oleic acid robustly increased p38 MAPK expression compared to the HS treatment. This observation suggests a potential role of oleic acid in modulating intracellular signaling pathways involved in myogenic differentiation. The p38 MAPK family, particularly the α, β, and γ isoforms, are well-documented for their involvement in the regulation of myogenesis, where they promote the activation of transcription factors such as *MyoD*, crucial for muscle-specific gene expression and differentiation [55]. Our results are consistent with previous findings, which demonstrated that the expression of p38 α, β, and γ isoforms is upregulated during the differentiation of C2C12 myoblasts [56] and supports the role of p38 MAPK signaling in promoting myogenic differentiation by enhancing the expression of myogenic regulatory factors [57].

Additionally, our results indicated that oleic acid significantly upregulated NDRG1, a protein implicated in differentiation and metabolism [58,59].

NDRG1 activation has been shown to promote myotube formation and enhance myogenic marker expression, including MyHC, during skeletal muscle differentiation [52]. Moreover, the *NDRG4* gene is upregulated during myogenic differentiation via Akt/CREB activation [60]. Our study supports this notion, as oleic acid-induced NDRG1 activation coincided with increased MyHC4 expression and multinucleated myotube formation.

Myoblast differentiation and fusion are also regulated by the Wnt/β-catenin signaling pathway, which governs the activation of myogenic genes. Previous studies have demonstrated that β-catenin is crucial for C2C12 differentiation, promoting morphological differentiation and gene program activation [61,62]. β-catenin is a protein involved in cell adhesion, acting as a key component of adhesion complexes in cell junctions. Furthermore, β-catenin also plays a significant role in myogenesis, influencing morphological differentiation and the activation and regulation of genes involved in muscle formation. The absence of β-catenin in primary myoblasts leads to the altered alignment, elongation, and fusion of cells, as well as a lack of coordination between myogenic gene expression and cytoskeletal and membrane remodeling events [63]. Our findings revealed significantly higher β-catenin expression in oleic acid-treated cells compared to HS-treated cells. Given the well-established role of β-catenin as a key effector in the Wnt signaling pathway, which is known to influence myogenic differentiation [64,65], this observation suggests that oleic acid may modulate molecular pathways associated with myogenesis. However, further studies are needed to elucidate the specific mechanisms involved.

Energy metabolism plays a crucial role in myoblast differentiation, with glucose uptake being a key process regulated by p38 MAPK. One of the downstream targets of p38 MAPK is GLUT4, an insulin-sensitive glucose transporter essential for glucose uptake in differentiated muscle cells [66]. Research indicates that the expression of GLUT4 increases as C2C12 myoblasts differentiate into myotubes, enhancing the cells’ capacity for insulin-stimulated glucose uptake [67]. Our study demonstrated that oleic acid significantly upregulated GLUT4 expression compared to the HS treatment. Furthermore, we observed a higher pIR/IR ratio in oleic acid-treated cells. Insulin receptor phosphorylation (pIR) at tyrosine residues is the initial step in propagating the insulin signal. Its increase in the oleic acid-treated group suggests improved insulin sensitivity. PDK1 is a critical kinase downstream of PI3K, essential for the full activation of AKT, and its phosphorylation at Ser241 is a hallmark of insulin pathway activation [68]. The increased pPDK1/PDK1 ratio further supports the activation of the insulin signaling pathway in response to oleic acid. Additionally, the phosphorylation of NDRG1, which serves as a downstream marker of nutrient and insulin signaling [69], is significantly upregulated. Notably, the ability of insulin to stimulate glucose uptake has been linked to the activation of p38 MAPK [70]. Therefore, our findings are consistent with these observations.

## 5. Conclusions

In summary, our findings reveal that oleic acid promotes C2C12 cell differentiation through molecular mechanisms distinct from those activated by the HS condition. By enhancing insulin signaling, evidenced by increased phosphorylation of key proteins (pPDK1, pIR), upregulation of GLUT4, and elevated pNDRG1, oleic acid appears to boost glucose uptake, thereby providing essential metabolic support for myogenesis. These results not only highlight the metabolic benefits of oleic acid but also suggest its potential as a serum-free alternative for promoting myoblast differentiation in vitro. Although our results suggest a potential therapeutic role for oleic acid in muscle regeneration, we acknowledge that this study was conducted in vitro using C2C12 cells. Therefore, we refrain from providing direct dietary recommendations. Nonetheless, this work contributes to a growing body of literature highlighting the bioactive properties of individual fatty acids and supports the notion that oleic acid may serve not only as an energy substrate but also as a modulator of anabolic signaling pathways relevant to muscle physiology. Further studies in animal models and clinical settings are needed to assess the bioavailability, effective dosages, and therapeutic potential of oleic acid or oleic acid-enriched diets in the context of muscle injury or degenerative conditions.

## Figures and Tables

**Figure 1 biomedicines-13-01568-f001:**
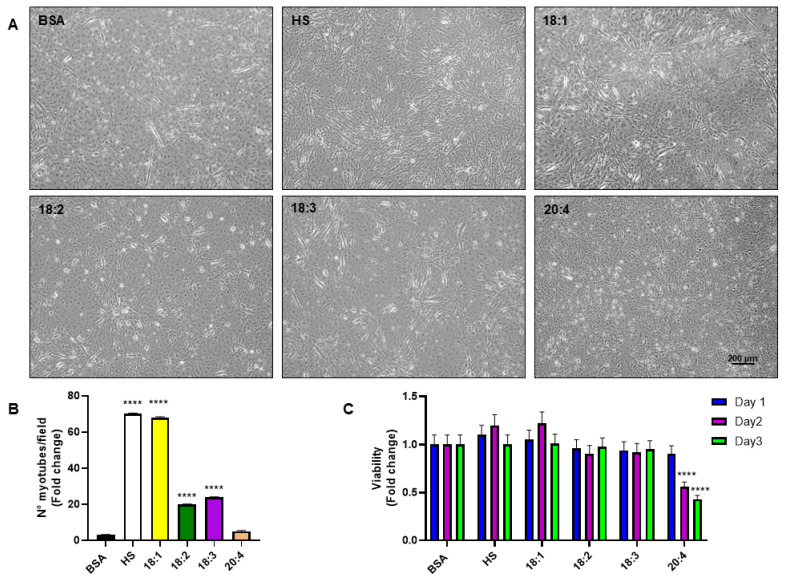
Effects of different fatty acids on C2C12 myotube formation. (**A**) Bright field microscopy images of C2C12 cells captured at days 3 post-differentiation induction. Cells were treated with 200 μM of various fatty acids, oleic (C18:1), linoleic (C18:2), linolenic (C18:3), and arachidonic acid (C20:4), or maintained under standard differentiation conditions with HS. Scale bars 200 μm. (**B**) Quantification of myotube formation at day 3 as the number of myotubes per microscopic field (expressed as fold change) under each treatment condition. (**C**) Assessment of cell viability across the three-day differentiation period. Values are means ± SE of three different experiments. (****) *p* < 0.001 versus BSA.

**Figure 2 biomedicines-13-01568-f002:**
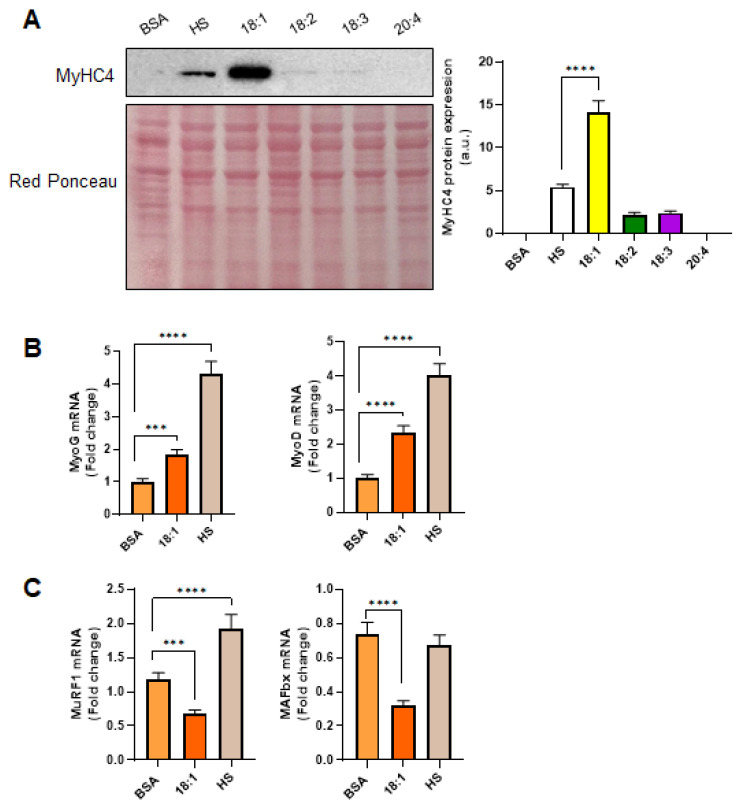
Oleic acid-induced C2C12 myogenic differentiation. (**A**) Western blot analysis of MyHC4 and relative quantification in C2C12 cells treated with BSA, HS, and 200 μM of oleic acid (C18:1), linoleic acid (C18:2), linolenic acid (C18:3), and arachidonic acid (C20:4). (**B**) mRNA level of myogenic markers *MyoD* and *MyoG*, and (**C**) of *Murf1* and *Mafbx* expressed as fold change of the control (BSA). Values are the means ± SE of three different experiments. (***) *p* < 0.005 versus BSA; (****) *p* < 0.001 versus BSA.

**Figure 3 biomedicines-13-01568-f003:**
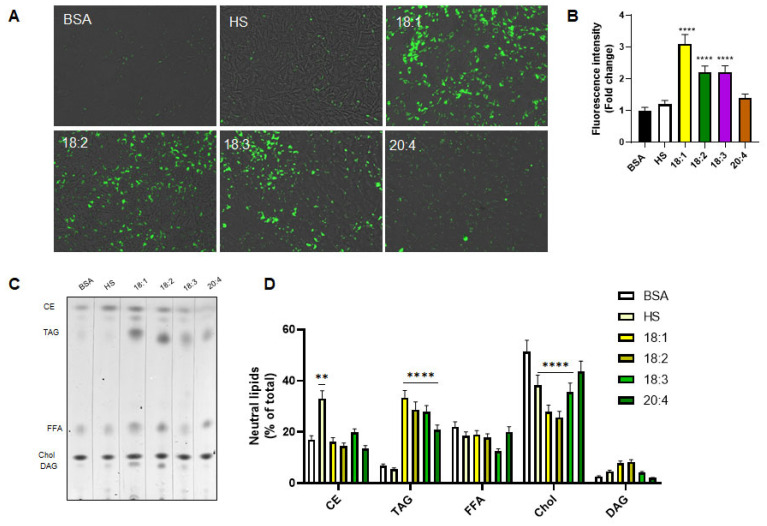
Fatty acid-dependent modulation of LD accumulation and intracellular lipid composition during C2C12 myogenic differentiation. (**A**) BODIPY staining of C2C12 cells after three days of differentiation in the presence of BSA, HS, and four different fatty acids. Scale bars 100 μm. (**B**) BODIPY-associated fluorescence measured by a plate reader scanner. (**C**) TLC analysis of intracellular lipid classes, including CEs, TAG, FFAs, cholesterol (Chol), and DAGs, following three days of treatment. (**D**) Quantification of lipid classes from TLC was performed by densitometric analysis. Each lipid species is expressed as a percentage of the total neutral lipid content and normalized to the control group (BSA). Values are the means ± SE of three different experiments. (**) *p* < 0.01; (****) *p* < 0.001 versus BSA.

**Figure 4 biomedicines-13-01568-f004:**
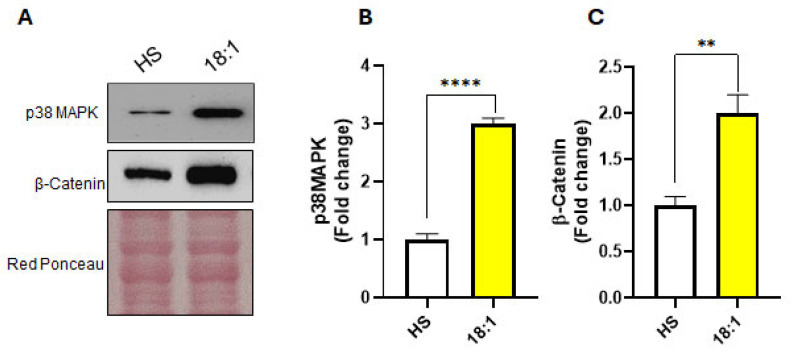
Oleic acid activates the p38MAPK/β-catenin pathway. (**A**) Representative Western blots of p38 MAPK and β-catenin in cells treated with HS or oleic acid (C18:1). (**B**,**C**) Quantitation of p38 MAPK and β-catenin expression in Western blots by densitometric analysis. The values were normalized to red ponceau. Protein amounts are expressed as fold change of HS-treated cells. The graph shows the means ± SE of three different experiments. (**) *p* < 0.01; (****) *p* < 0.001 versus HS.

**Figure 5 biomedicines-13-01568-f005:**
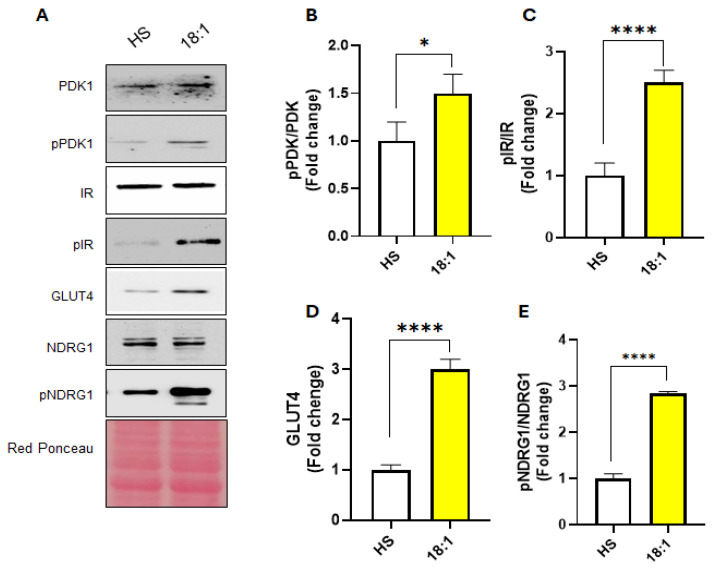
Oleic acid activates the insulin signaling. (**A**) Representative Western blots of PDK1, pPDK1, IR, pIR, GLUT4, NDRG1, and pNDRG1 in cells treated for three days with HS or oleic acid (C18:1). (**B**–**E**) Quantitation of pPDK1/PDK1, pIR/IR, GLUT4, and pNDRG1/NDRG1 expression in Western blots by densitometric analysis. The values were normalized to red ponceau. Protein amounts are expressed as fold change of HS-treated cells. The graph shows the means ± SE of three different experiments. (*) *p* < 0.05; (****) *p* < 0.001 versus HS.

## Data Availability

The original contributions presented in this study are included in the article/Appendix A. Further inquiries can be directed to the corresponding author.

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
