# Peer review of "Exploring the Role of Oleic Acid in Muscle Cell Differentiation: Mechanisms and Implications for Myogenesis and Metabolic Regulation in C2C12 Myoblasts"

_biomedicines, 2025, doi:10.3390/biomedicines13071568_

Round 1
Reviewer 1 Report
Comments and Suggestions for Authors
Dear Authors.
Dear authors the paper presented for review is interesting and deals with the important issue of the activity of fatty acids, and their properties in the regeneration of muscle tissue. After reading the paper, I have the following comments.
1. the chapter on materials and methods should be distinguished by a separate chapter on cell cultures, their preparation for the study, the origin of the cell line and the specific conditions of culture. Why these and not other cells were chosen. Is this a validated model for muscle regeneration in humans. Necessary extensive justification with literature and description also at the beginning of the chapter discussion.
2. the work also often lacks reference to the manufacturer and origin of the materials, reagents and equipment used. This should be supplemented.
3.The paper should also describe in detail how the fatty acids were added to the culture medium (e.g., whether they were in the form of an emulsion), since they themselves are insoluble in water and, if the experiments were performed as described, they would be on the surface and would not affect the cells. According to the description, the work was not done correctly and is not suitable for printing. I hope this is an error in the description.
4. on what instrument the absorbance was read, how were the cells counted for the test, All descriptions in must be accurate. You should describe all steps point by point. This applies to all descriptions of methods.
5, What was used microscope, magnification, software for readings.
6, Microscopic images are not legible, it is suggested to limit their number and transfer part to the supplement. Lack of scale and proper description under the images.
7. under the graphs should be supplemented with information on statistical significance, e.g. Fig. 2.
8. conclusions, what conclusions. The reader may have the impression that the work was about something else. Why was the research done and what does it contribute. If I damage a muscle, should I drink oil, possibly what kind? How much (oil is emulsified after ingestion - such a note to the methodology).
9. Please add a graphic summary of the results.
Author Response
Reviewer 1:
Dear authors the paper presented for review is interesting and deals with the important issue of the activity of fatty acids, and their properties in the regeneration of muscle tissue. After reading the paper, I have the following comments.
- the chapter on materials and methods should be distinguished by a separate chapter on cell cultures, their preparation for the study, the origin of the cell line and the specific conditions of culture. Why these and not other cells were chosen. Is this a validated model for muscle regeneration in humans. Necessary extensive justification with literature and description also at the beginning of the chapter discussion.
- We thank the reviewer for your valuable suggestion. In response, we have expanded the Materials and Methods section to include detailed information on the origin of the cell line and the specific culture conditions. Furthermore, we have added a dedicated paragraph in the Discussion section to justify our choice of the C2C12 cell line, supported by relevant literature evidence.
- The work also often lacks reference to the manufacturer and origin of the materials, reagents and equipment used. This should be supplemented.
- We thank the reviewer for your comment. We have updated the Materials and Methods section to include detailed information on the materials, reagents, and equipment used, as requested.
3.The paper should also describe in detail how the fatty acids were added to the culture medium (e.g., whether they were in the form of an emulsion), since they themselves are insoluble in water and, if the experiments were performed as described, they would be on the surface and would not affect the cells. According to the description, the work was not done correctly and is not suitable for printing. I hope this is an error in the description.
- We thank the Reviewer for pointing out the need for clarification on this methodological aspect. We fully agree that the preparation and delivery method of fatty acids is critical to ensure their bioavailability and biological relevance in in vitro systems. In response to your concern, we have revised the Methods section to clearly describe that fatty acids were first conjugated to fatty-acid-free bovine serum albumin (BSA) at a molar ratio of 2:1 (FA:BSA), following established protocols [https://doi.org/10.1016/j.bbalip.2017.11.006]. This approach not only prevents fatty acid precipitation but also mimics physiological transport conditions, as albumin is the primary fatty acid carrier in plasma. We already used this protocols in our recent published researches (doi: 10.1016/j.jlr.2024.100692). We appreciate the opportunity to clarify this key methodological aspect and believe the updated description resolves the concern about the validity and reproducibility of our experimental setup.
- On what instrument the absorbance was read, how were the cells counted for the test, All descriptions in must be accurate. You should describe all steps point by point. This applies to all descriptions of methods.
- Dear reviewer, we have further investigated what you requested, we have provided more information about the quantity of myotubes formed following differentiation, in paragraph 2.2.
5, What was used microscope, magnification, software for readings.
- We thank the reviewer for the appoint. In response, we have implemented the information about microscopy images in the Materials and Methods section.
6, Microscopic images are not legible, it is suggested to limit their number and transfer part to the supplement. Lack of scale and proper description under the images.
- We appreciate the Reviewer’s attention to detail and have addressed the issue accordingly. Since Figure 1 includes not only the microscopic images of cells treated with various fatty acids, but also data on cell viability and myotube formation, we decided to retain all microscopy panels in the main figure to preserve the overall context. However, to improve clarity, we have added enlarged versions of selected images, specifically for BSA, HS, and oleic acid treatments, in the Figure S1, where the morphology of the myotubes is more clearly visible. Scale bars and detailed figure legends have been added to ensure proper interpretation and context. We hope these revisions adequately address the Reviewer’s concerns.
- under the graphs should be supplemented with information on statistical significance, e.g. Fig. 2.
- We thank the Reviewer for highlighting this point. In response, we have revised the caption of Figure 2 (and other relevant figures, where appropriate) to include detailed information regarding statistical significance. We appreciate your suggestion, which has helped us improve the clarity and transparency of our data presentation.
- conclusions, what conclusions. The reader may have the impression that the work was about something else. Why was the research done and what does it contribute. If I damage a muscle, should I drink oil, possibly what kind? How much (oil is emulsified after ingestion - such a note to the methodology).
- We thank the reviewer for the insightful comments. We agree that the original conclusion section could benefit from greater clarity and emphasis on the study’s motivation and implications. In response, we have revised the Conclusions section to explicitly address the rationale behind the study, the main findings, and their potential relevance to muscle health and regenerative strategies. We would like to clarify that our study is an in vitro investigation using a controlled cell culture model. While oleic acid is a major component of commonly consumed oils such as olive oil, our data do not directly support clinical or nutritional recommendations for fatty acid intake. As noted in the revised Methods section, oleic acid was conjugated to bovine serum albumin (BSA) to simulate physiological fatty acid transport in the bloodstream and to ensure solubility and bioavailability in the culture medium; this does not reflect how oleic acid behaves during human digestion, where emulsification occurs via bile salts. We have included a clarification in the revised Methods to emphasize this point and prevent misinterpretation. While promising, our findings should be considered a foundational step that warrants further validation in in vivo models before any translational or dietary conclusions can be drawn.
- Please add a graphic summary of the results.
- As suggested by the reviewer, a graphical abstract highlighting the key findings of the study has been added to the revised version of the manuscript
Reviewer 2 Report
Comments and Suggestions for Authors
This study provides compelling evidence that oleic acid not only promotes myogenic differentiation but also exerts protective effects against muscle atrophy. In contrast, horse serum (HS) appears to induce a broad activation of both anabolic and catabolic pathways. The results highlighting these differences, along with the authors’ interpretation, are highly interesting and offer valuable insights into the distinct biological actions of oleic acid versus serum-based differentiation.
Please refer to the following for a summary of the issues and corrections that need to be made to this paper.
<1>The authors state that arachidonic acid (20:4) resulted in minimal lipid droplet accumulation, suggesting that it may be preferentially utilized for eicosanoid synthesis or other metabolic pathways rather than storage (Figures 3A, B). However, could this observation also be attributed to potential cytotoxic effects of arachidonic acid?
<2>The legend in Figure 3 contains many obvious careless mistakes and needs to be corrected.
<3>The manuscript discusses the phosphorylation of p38 MAPK based on Western blot analysis; however, it should be noted that the antibody used (Cell Signaling #9212) does not detect the phosphorylated form of p38 MAPK. The correct antibody for detecting phosphorylated p38 MAPK is Cell Signaling #9211. Therefore, the current experimental data do not support conclusions regarding changes in p38 MAPK phosphorylation levels. 
<4> As markers of fast-twitch muscle fibers, Myh4, Myh1, and Myh2 are commonly used. If the authors intend to discuss gene expression related to fast fiber type specification, it is recommended that they include analysis of these genes to strengthen their conclusions. Given that cDNA has already been synthesized, assessing the expression levels of these genes would be a straightforward experiment that could be completed within a few days.
<5> It is unclear what is specifically meant by "high serum conditions" in the conclusion. For clarity, the authors should define this term explicitly—does it refer to the use of 2% horse serum (HS) during differentiation, or a different condition? 
<6> The sequence NC_000067.7 corresponds to Mus musculus strain C57BL/6J chromosome 1, GRCm39. However, the primer sequences listed in the manuscript do not match the Myog mRNA sequence and therefore do not appear to be appropriate for amplifying Myog transcripts. The authors should confirm and clearly demonstrate that the primers used are specific and suitable for detecting Myog expression. This discrepancy could have a substantial impact on both the experimental results and the interpretation of the data, and thus must be carefully addressed.
<7> Please ensure that the primer set used for Mafbx amplification is appropriate. According to sequence analysis, the forward primer appears to perfectly match the target sequence; however, the reverse primer shows a 5-base mismatch, raising concerns about the specificity and efficiency of amplification. Additionally, Primer-BLAST analysis suggests that the expected amplicon size is 182 bp, while Table S1 lists it as 181 bp. These discrepancies should be addressed, as they may impact the accuracy and reproducibility of the gene expression results.
<8>The images in Fig. 1A, while useful for providing an overview due to their low magnification, do not clearly show myotube formation. To enable a more accurate assessment of myogenic differentiation, I recommend including higher-magnification images or implementing other strategies that better highlight myotube morphology.
<9> The use of the term "vitality test" in section 2.3 may need reconsideration, as "viability test" is the more commonly accepted terminology for assessing cell survival.
Comments on the Quality of English LanguageThe manuscript is clearly written, and the English language is appropriate and scientifically sound throughout. No major grammatical or stylistic issues were noted.
Round 2
Reviewer 1 Report
Comments and Suggestions for Authors
Dear authors.
Thank you for the corrections and answers to our questions.
1. Figure 1 is still unclear and needs to be corrected. Alternatively, please move the cell photographs to the supplement as they are unclear in this form, or try to improve the contrast.
2. Some of the descriptions of the reagents are incomplete or incorrect. For example, Agilent is not an Italian company and the device is manufactured in the USA.
Author Response
Dear Reviewer,
We sincerely thank you for your thorough and constructive feedback, which has greatly helped us improve the quality of our manuscript.
- Figure 1 is still unclear and needs to be corrected. Alternatively, please move the cell photographs to the supplement as they are unclear in this form, or try to improve the contrast.
- We have thoroughly addressed the concerns regarding Figure 1 by enhancing the image contrast to ensure improved clarity and visibility. To further optimize the presentation, we have focused Figure 1 exclusively on the 3-day time point, allowing for a larger, more detailed image. The complete time-course data for days 1, 2, and 3 have been relocated to Supplementary Figure S1 for comprehensive reference. Additionally, we have provided enlarged images of the BSA, HS, and oleic acid treatments in Supplementary Figure S2 to facilitate better visualization of the differences. All corresponding updates have been made to the main text and figure legends to accurately reflect these modifications.
- Some of the descriptions of the reagents are incomplete or incorrect. For example, Agilent is not an Italian company and the device is manufactured in the USA.
- We have carefully reviewed and corrected the descriptions of all reagents and equipment. In particular, we have updated the information regarding Agilent Technologies, specifying that it is a U.S.-based company, and we have verified and standardized the origin and manufacturer details for all listed products.
We truly appreciate your detailed review and valuable suggestions, which have significantly strengthened the manuscript.
Reviewer 2 Report
Comments and Suggestions for Authors
I fully understand all my comments and have ensured that the manuscript has been properly revised. I hope that the many careless mistakes made in the initial submission will be remedied in future submissions.
Author Response
Thank you very much for your thoughtful and constructive comments. We truly appreciate the time and effort you invested in reviewing our work, as your insights have been invaluable in helping us improve the quality and clarity of our manuscript.